# Lignin-Based Composite Film and Its Application for Agricultural Mulching

**DOI:** 10.3390/polym16172488

**Published:** 2024-08-31

**Authors:** Zujian Huang, Yan Zhang, Chenwei Zhang, Fangting Yuan, Hairong Gao, Qiang Li

**Affiliations:** 1College of Engineering, Huazhong Agricultural University, Wuhan 430070, China; zujianhuang0603@163.com (Z.H.); zyy0623@webmail.hzau.edu.cn (Y.Z.); 18840813195@163.com (C.Z.); 2College of Horticulture & Forestry Science, Huazhong Agricultural University, Wuhan 430070, China; ftyuan@mail.hzau.edu.cn

**Keywords:** lignin, composite material, mulching, biodegradability, agriculture practice

## Abstract

Agricultural mulching is an important input for modern agricultural production and plays an important role in guaranteeing food security worldwide. At present, polyethylene (PE) mulching is still commonly used in agricultural production in most countries around the world, which is non-biodegradable, and years of mulching have caused serious agricultural white pollution. Lignin is one of the three major components of plant cell walls, and it is also the main renewable natural aromatic compounds in nature. Lignin-based composite film materials are green, biodegradable, and show good prospects for development in the field of agricultural mulch. This paper introduces the types, structure, and application status of lignin, summarizes the preparation of lignin-based composite film materials and its latest research progress, focuses on the types, preparation methods, and application examples of lignin-based agricultural mulching, and looks forward to the future development prospects of lignin-based agricultural mulching.

## 1. Introduction

Since the first utilization of paper sheets as mulch in agriculture in the 1920s, mulching has been gradually used as an important agriculture practice worldwide [1]. In recent decades, modern agriculture has widely used mulching to improve grain yield and the quality of horticulture production, especially for agriculture in cold and arid areas [2]. Mulching can effectively improve the insulation and moisture retention of soil. By regulating the temperature of the soil and reducing the evaporation of water from the soil, it promotes plant growth, which is especially crucial during the cold season or in dry areas [3]. At the same time, as the temperature and moisture content of the soil increase, the number and activity of microorganisms in the soil could be increased to some extent. Such augmented biodiversity can enhance the effectiveness of nutrient uptake by the plant root system, and thus promote plant growth [4,5]. In addition, mulching also inhibits the growth of weeds, thereby reducing their competition with crops for water and nutrients [6].

As reported, the global mulching market reached USD 3.5 billion in 2020 [1]. Most agricultural mulches used nowadays are manufactured from petroleum derivatives (polyethylene, PE), which are known to be low-cost, easy to process, and lightweight [7]. However, PE mulch is non-biodegradable, resulting in the accumulation of residual mulch in the soil that can cause serious white pollution.

In general, white pollution refers to the pollution induced by waste plastic products, including disposable tableware, plastic bottles, and plastic mulch films, all of which are indiscriminately discarded as solid waste after use, thereby causing pollution to the ecosystem and landscape as they are very difficult to be degraded and disposed of [8]. For modern agriculture, white pollution has become a potential threat. Plastic mulch left in the environment may break down into microplastics (MPs), the presence of which may alter the chemical and physical properties of the soil, destroying its original structure [9]. It may also affect the physiology of plants, including root hair development, root biomass, and leaf area, which may eventually impact crop yield and quality [10]. In addition, the small size and remarkable adsorption of MPs can lead to their entrance into plants and animals, which may then be ingested by people through a variety of dietary sources, posing potential risks to human health [10,11].

Lignocellulose, such as forestry and agricultural residues, represents the most abundant natural resources on the earth, with the advantages of renewability and biodegradability [12]. This lignocellulosic biomass mainly comprises three biopolymers: cellulose, hemicellulose, and lignin [13]. In both the lignocellulosic biorefinery and the pulp and paper industries, cellulose is already extracted and hydrolyzed into fermentable sugars or converted into commercially valuable products, while lignin is usually disposed of as a waste stream [14,15]. Because of this, lignin constitutes a cost-effective, readily accessible renewable resource that could be utilized more efficiently for the production of biological products, in contrast to cellulose, keratin, proteins, and other natural biopolymers. Furthermore, lignin itself can serve as the soil organic matter, enriching field fertility after being used as agricultural mulching. Because of these advantages, lignin has great potential among different biopolymers for biodegradable mulching.

However, the complex structure and poor processability of lignin make its utilization difficult. Also, it is a major waste product from the paper industry and biorefining processes. In recent years, the high-value utilization of lignin has become a research hotspot, which is used to prepare a variety of high-value-added products through the directional modulation of lignin structure [16]. Because of its excellent water resistance, UV resistance, and biodegradability, lignin has been used as a functional additive for green mulch materials to strengthen the mechanical strength and optical properties of plastic mulch [17]. Starting from the basic structure, separation, and deconstruction of lignin, this review gives a systematic summary of the types, preparation methods, and properties of lignin composite film materials, focusing on the current status of the application of lignin in agricultural mulch to provide support for the application of biodegradable mulch materials based on woody biomass in agriculture.

## 2. Overview of Lignin

### 2.1. Structural Units and Linkages of Lignin

Lignin is a natural polymer formed by linking phenylpropane building blocks through C-C bonds or C-O-C ether bonds. As shown in Figure 1, according to the difference in the number of methoxy groups on the aromatic ring, lignin is mainly categorized into three types: *p*-Hydroxyphenyl unit (H), Guaiacyl unit (G), and Syringyl unit (S) [18]. The type and content of lignin in different plant fiber raw materials vary [19], for example, the lignin type in coniferous wood is relatively homogeneous and is dominated by the G-type, the lignin in broadleaf wood is dominated by the G-type and the S-type, whereas the lignin in gramineous crops contains a certain amount of the H-type in addition to the G-type and the S-type.

The linkages between individual structural units in lignin mainly include C−C bond type connections such as *β*-5, *β*-1, and 5-5, and ether bond type connections such as *β*-O-4, *α*-O-4, and 5-O-4 (Figure 1), with a certain number of ester-bonded linkages in individual gramineous plants [19]. Typically, ether bond linkages account for 60% to 70% and C−C bond linkages account for 30% to 40% [20]. In addition, lignin molecules contain a variety of functional groups such as methoxy (-OCH_3_), phenolic hydroxyl (Ar-OH), aliphatic hydroxyl (Alk-OH), and side-chain carbonyl (C=O) on the benzene ring. These functional groups endow lignin with excellent UV, antioxidant, antimicrobial, and flame-retardant properties, making lignin widely used in packaging, chemical, and agricultural applications [21,22].

The structure of lignin is usually highly heterogeneous, with significant differences in molecular weight size, type, number of functional groups, and intramolecular connections among different lignin, making it one of the most structurally complex macromolecules in nature. In addition, scholars generally recognize that lignin is a barrier to the dissociation of plant cell wall structure, and the development of green and efficient delignification technology is the key to the sustainable development of the pulp and paper industry and the biorefining industry.

### 2.2. Lignin Separation Techniques

The separation of lignin from the cell wall is one of the central aspects of the industrial utilization of existing woody biomass. During the separation process, the lignin itself and the connections it forms with natural polysaccharides such as cellulose and hemicellulose will be broken, resulting in the separation of the lignin fraction from the polysaccharide fraction [23]. Lignin separation methods are diverse, the typical methods include milled wood lignin (MWL), Kraft lignin, lignosulfonate, and so on. Different types of lignin have obvious structural differences due to their different preparation methods (Figure 2) and are therefore suitable for applications in different fields [24].

First, as shown in Figure 2a, MWL is a type of lignin prepared by physical ball milling of biomass feedstock and then extracted with organic reagents, which is often used as a representative of natural lignin due to the relatively small damage to the structure of lignin in the preparation process and is usually considered to be similar to the natural structure of lignin in biomass [24]. In the MWL preparation process, ball milling can physically deconstruct cell walls, and the lignin in the cell wall is subsequently solubilized with 1,4-dioxocyclohexane organic reagent, and then MWL can be prepared by purification and drying [25,26]. Yang et al. [26] prepared MWL from eucalyptus and measured the dry and wet bond strength of triple-layer plywood prepared with them by the lignin can be used to prepare high-strength lignin adhesives with a simple and efficient method of operation, which is “green” and environmental-friendly.

Second, Kraft lignin is produced by reacting plant materials with sodium hydroxide and non-hydrated sodium sulfide (Figure 2b) [27,28]. In general, Kraft lignin is a highly modified hydrophobic lignin with lower molecular weight and sulfur content than the original lignin. In addition, Kraft lignin is generally pretreated with modifications by sulfonation or amine reactions before being used as a dispersant [15].

Thirdly, lignosulfonate is a by-product of sulfite pulping, which is produced by reacting plant raw materials with sulfurous acid (magnesium, calcium, sodium, or ammonium-based) at different pH values. The sulfite delignification process generally precipitates lignin by the acidification of the waste stream method, during which the lignin molecule undergoes acidic cleavage of the ether bond, and the resulting electrophilic carbon ions will react with sulfite ions to form sulfur-containing groups (Figure 2c) [29,30]. Compared to lignin sulfates, sulfonated lignin is usually more sulfated [31].

By comparing the three different methods for lignin separation, it can be seen that MWL reduces the size of the lignocellulosic particles, increases fiber accessibility, and allows lignin to be extracted more easily. However, the structure of the lignin obtained by this method varies depending on the milling time and intensity [32]. Kraft lignin is the dissolved lignin component of the main technology used to produce unbleached chemical pulp and can be well-suited for value-added production. However, limitations in its use are caused by its heterogeneity. It exhibits non-homogeneity not only in structure but also in chemical function and molecular weight [28]. Moreover, lignosulfonates account for 90% of the total commercial lignin market. The fact that lignosulfonates are anionic and water-soluble makes them very versatile [30].

**Figure 2 polymers-16-02488-f002:**
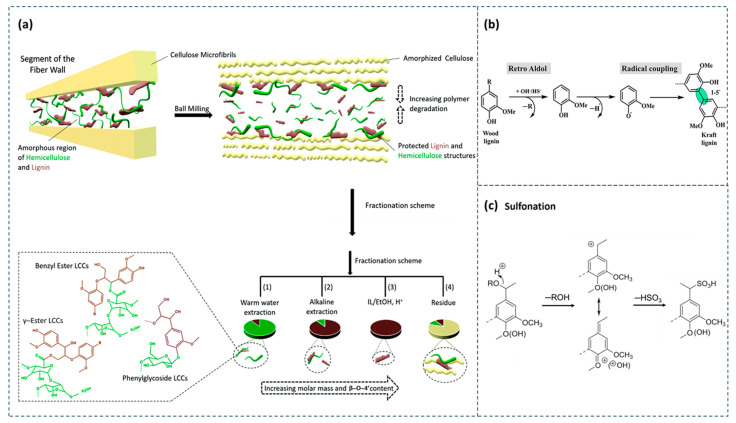
Processes of three different lignin preparation methods. (**a**) Destination of different components inside the cell wall during MWL preparation [32]. (**b**) Preparation process of Kraft lignin [28]. (**c**) Preparation process of lignosulfonate [30].

## 3. Lignin-Based Composite Film Materials

Due to the biodegradability and biocompatibility of lignin molecules, they are widely used in composites as fillers, compatibilizers, coupling agents, etc. The direct addition of lignin may reduce the mechanical strength or other properties of the material [33], so the modified pretreatment of lignin can significantly improve the mixing and co-compatibility of lignin and other raw materials. Lignin-based composite film materials are generally categorized into two main groups based on raw materials: lignin/natural polymer composite film and lignin/synthetic polymer composite film. Natural polymers are further divided into plant-based polymers, animal-based polymers, and microbial-based polymers, which are widely available and inexpensive (Figure 3).

### 3.1. Lignin/Natural Polymer Composite Film

Natural polymer materials, including cellulose, starch, and chitosan have good biocompatibility and a variety of functional properties and can be compounded with lignin to prepare new environmentally friendly film materials with high added value. The following will introduce several representative composite film materials.

#### 3.1.1. Lignin/Cellulose Composite Film

Cellulose is the most abundant natural polymer in nature, as the main component of plant cell walls, it can be obtained from straw, forestry waste, and other waste resources. It consists of linear polymer chains of anhydrous *D*-glucopyranoside units with a large number of inter- and intramolecular hydrogen bonds formed [34,35]. Most of the hydrophilic groups of lignin are wrapped in hydrophobic chains, making them incompatible with the hydrophilic functional groups in cellulose, while both cellulose and lignin are negatively charged under general conditions, leading to electrostatic repulsion between them [36].

Currently, a variety of preparation techniques based on mechanical methods (e.g., milling, high-pressure homogenization, ultrasonic treatment), chemical treatments (e.g., conventional pulping process, inorganic acid hydrolysis, enzymatic hydrolysis), and a mixture of both have been developed for the preparation of lignin/cellulose composite film [37] (Figure 4). In order to achieve a “greener” and more efficient preparation process, more and more studies have been conducted to prepare lignin-cellulose composite film using “green” solvents. For example, ionic liquids (ILs), deep eutectic solvents (DES), etc., are simpler, greener, and have safer reaction conditions compared to traditional preparation methods such as DMSO/water and dioxane/water systems [38]. Ionic liquids as co-solvents can promote the realization of lignin and cellulose co-mingling, and the formation of strong hydrogen bonds between the two with water, which in turn makes the film have excellent properties. Colburn et al. [39] used the ionic liquid (1-ethyl-3-methylimidazole) acetate as the solvent of microcrystalline cellulose, and at the same time, lignosulfonate was uniformly dispersed in the suspension, and the film was prepared by casting method to obtain a film with good antifouling properties and good resistance barrier.

#### 3.1.2. Lignin/Starch Composite Film

Starch is a polymer commonly found in nature, which is not only one of the most widely used natural raw materials but also used in a variety of applications due to its renewable and biodegradable properties. The abundance of hydroxyl groups in the starch molecule gives it a high degree of hydrophilicity, a property that makes starch-based film materials highly hygroscopic, leading to structural changes that affect their mechanical properties. This weakness seriously limits the wide application of starch-based materials in packaging, agricultural mulch, and the medical industry, especially in the fields where it is desired to replace traditional plastics in order to reduce environmental pollution. This has become a problem that needs to be urgently solved.

Lignin/starch composite film materials can be a good solution to the problems of poor mechanical properties, moisture sensitivity, and critical aging of starch-based materials themselves [40]. Lignin and starch can be used to prepare films by thermoforming or casting without the aid of cross-linking agents. Alternatively, the lignin and starch materials can be mixed in a mixer to form a casting blend with the addition of certain plasticizers (e.g., glycerol) and then use a screw apparatus to extrude it into shape or cast it onto a mold and dry it to form it. Baumberger et al. [41] in 1997 first proposed that the addition of glycerol improves the properties of wheat starch and lignosulfonate composite films, and the mechanical properties of the composite films are greatly improved compared to starch-only films.

Nanosized lignin can be used as a hydrophobic modifier for composite films. Lignin nanoparticles were prepared by mixing lignin with tetrahydrofuran using the anti-solvent method, and then they were added to starch and polyvinyl alcohol solutions to produce thin films by the casting method [42]. The hydrophobic methoxy groups in lignin combined with starch to form a tight hydrophobic layer, starch and lignin formed strong hydrogen bonds in the presence of polyvinyl alcohol.

In addition, the lignin-starch composite film has excellent thermal stability and UV protection. Wu et al. [43] also prepared ionic liquids with 1-allyl-3-methylimidazolium chloride amine and dispersed lignin, starch, and cellulose in a certain proportion to which the blended film was obtained by casting method. The composite film has good thermal stability and mechanical properties, and the ionic liquid can be better recycled, which has a broad prospect in the field of fresh food packaging.

#### 3.1.3. Lignin/Chitosan Composite Film

Chitosan is a kind of material extracted from the shells of shrimps and crabs with biodegradability and good biocompatibility [44]. Chitosan polymer surface contains abundant amino and hydroxyl groups, allowing it to combine with a large number of carboxyl and hydroxyl functional groups in lignin macromolecules through hydrogen bonding and van der Waals forces. This interaction helps achieve the requirements for mechanical properties and thermal stability of composite film [45], which are usually prepared by simple blending or film casting methods to combine the two to prepare composite film materials.

The free phenolic hydroxyl groups in lignin can confer antioxidant function to the film materials, as well as improve the antimicrobial activity of the material [46]. Shivani et al. [47] prepared novel chitosan/lignin film materials by solution casting method, which homogenized the lignin by NaOH, changed the hydrophobicity of the lignin itself, which was further used as a film materials adsorbent, and by changing the proportion of the lignin was finally able to achieve 95% removal of methylene blue dye. Ji et al. [46] prepared biodegradable chitosan-based films containing micro ramie fibers and lignin by casting method. By regulating the ratio of fibers and lignin (both micro ramie fibers and lignin were added at a ratio of 20%), the film material has excellent tensile properties, oxidation resistance, and water resistance, and it can be used in the packaging of fresh food.

#### 3.1.4. Lignin/Bacterial Cellulose Composite Film

Bacterial cellulose is a natural polymer material produced by bacteria such as *Acetobacter xylinum* [48]. Bacterial cellulose aggregates to form ribbon-like filaments and gradually forms a unique porous three-dimensional network structure. Compared to plant cellulose, bacterial cellulose has higher purity, crystallinity, and mechanical strength [49]. Bacterial cellulose has been widely used in several applications due to its unique nanoscale fiber network structure, including as an electrolyte carrier for supercapacitors and batteries, as a backbone for reinforced composites, and as a reinforcing material in textile and paper manufacturing.

The addition of lignin fills the gaps between the fibers within the bacterial cellulose and improves the film’s waterproof, tensile, oxidation, and UV protection properties. Dai et al. [50] obtained the lignin by deep eutectic solution (DES) isolation prepared from choline chloride and lactic acid. The lignin extracted by this method is rich in phenolic hydroxyl groups and low molecular weight, which enables the formation of stable hydrogen bonds between lignin and bacterial cellulose, resulting in films with excellent mechanical strength, UV resistance, and antioxidant properties (Figure 5).

Moreover, the addition of lignin can also delay the biodegradation cycle of the whole biomass-based bacterial cellulose films, which can further broaden the application areas. Tian et al. [51] prepared lignin nanoparticles by three different processes: deep eutectic solvent, ethanol organic solvent, and soda ash/anthraquinone, respectively. The lignin/bacterial cellulose nanocomposites obtained showed better-delayed degradation performance, which was more significant under the condition of higher enzyme concentration.

### 3.2. Lignin/Synthetic Polymer Composite Film

Synthetic polymers prepared film materials with excellent mechanical properties, but most of them do not have antibacterial, antioxidant, and anti-ultraviolet functions, which makes it difficult to be directly applied on a large scale in the agricultural mulch and food preservation field. The addition of lignin can significantly improve the antimicrobial, antioxidant, and UV-resistant properties of film materials, and in addition, the addition of lignin can reduce the cost of film materials to a certain extent [52]. In lignin/synthetic polymer composite film materials, lignin can be blended with synthetic polymer materials such as polyolefin, polyvinyl alcohol, and polyvinyl chloride.

#### 3.2.1. Lignin/Polyolefin Composite Film

Polyolefin is a class of polymerization monomers for olefin (e.g., polyethylene (PE), polypropylene (PP), etc.) polymer. Polyolefin composite films have excellent physical and chemical properties, including good chemical resistance, abrasion resistance, barrier properties, and mechanical strength. The blending of PE and PP with lignin improves the adhesion of bio-based media and is also able to impart the antimicrobial properties of lignin to the blends. However, lignin is a strong polar material with hydrogen bonding, which makes the blending between the two less effective [53], so the compatibility of lignin is generally increased by chemical (e.g., reactions such as esterification, etherification, grafting and copolymerization) or plasticization modifications, or by the addition of capacitance-enhancing agents (e.g., epoxy-containing capacitance-enhancers) to improve the adsorbent power of lignin. Cazacu et al. [54] used epoxy-modified lignosulfonate blended with polyolefin copolymer (70% polyethylene and 30% polypropylene) in the presence of a bulking agent, Exxelor 805, to produce films with good thermal, physical-mechanical, and good surface free energies by hot pressing. At low concentrations, modified lignosulfonates were able to be rapidly dispersed into the polyolefin blends, further improving the mechanical properties of the film materials. However, the thermal stability and mechanical properties of the blends decreased to varying degrees with increasing concentration and the films reached their optimum performance at a concentration of 10%. Chiappero et al. [55] proposed the preparation of composite films by blending unmodified softwood lignin and esterification-modified hardwood lignin with linear low-density polyethylene via twin-screw melt blending. They found that both unmodified softwood lignin and esterified hardwood lignin increased the stiffness of the polymer. It was found that the addition of unmodified lignin increased the stiffness of the polymer at the expense of its tensile strength and elongation at break. The decrease in polarity and molecular weight of lignin after esterification reduces its interaction with polyolefin, enhancing their compatibility. This compatibility enhancement leads to film materials with good heat resistance, mechanical properties, water absorption, and biodegradability.

#### 3.2.2. Lignin/Polyvinyl Alcohol (PVA) Composite Film

PVA is currently the only polyethylene-based synthetic polymer that is biodegradable [56]. Under natural conditions, biodegradation of polyvinyl alcohol is slow. However, the addition of appropriate lignin to polyethylene can effectively improve the biodegradability and mechanical strength of the polymer. Since lignin has a large number of polar functional groups that can form strong hydrogen bonds with the hydroxyl groups in PVA, the two have good compatibility. More and more studies have been conducted to prepare films by nanosizing lignin and then dispersing it into polyvinyl alcohol. Xu et al. [57] dissolved lignin in *γ*-valerolactone/aqueous solution, dispersed the lignin nanoparticles into polyvinyl alcohol after dialysis precipitation method, and then prepared the films by casting method (Figure 6). It was found that the addition of lignin nanoparticles to polyvinyl alcohol could effectively improve the thermal stability and UV shielding properties of the films, presenting better biodegradability, which is favorable for the development and application in the fields of biodegradable food packaging films and medical packaging materials.

Similar to the above-mentioned polymers, lignin has been also composited with other natural biopolymers derived from microbial fermentation like PLA and PHA as well as synthetic polymers like PBAT (Figure 3). As shown in Table 1, typical lignin/PLA film has tensile strength of 55.1 MPa and Young’s modulus of 1589 MPa, lignin/PBAT has tensile strength of 30 MPa, Young’s modulus of 63 MPa, and elongation at break of 689%, and lignin/PHA composite film has tensile strength of 82 MPa and Young’s modulus of 827 MPa. All these mechanical performances have been enhanced greatly by compositing lignin with these guest polymers.

## 4. Application of Lignin-Based Composite Film in Agriculture

### 4.1. Agricultural Practice with Mulching

Agricultural mulching film has become the fourth largest agricultural product after seeds, pesticides, and fertilizers with a huge application area and unique application mode [61]. Mulching can provide a good growing environment for crops and at the same time can reduce the impact of crop damage caused by natural disasters.

As shown in Figure 7, mulching is conducive to regulating soil temperature, increasing nutrient absorption, and accelerating plant germination and growth. It also controls weeds and reduces the incidence of diseases and salt stress, reduces soil compaction and erosion, and increases crop yield [62]. Organic mulch aids in nutrient management by releasing essential plant nutrients and organic carbon upon gradual decomposition by microbes [63]. In addition, changes in the ecological microenvironment of the farmland as well as the temperature and water storage in the soil by mulching will increase the activity of enzymes, such as urease and catalase in the soil [64], which play an important role in the growth and development of the crops and are conducive to increasing the activity of the root system, promoting the growth of the root system, and accelerating the reproductive process of the crops. Meanwhile, the root growth of crops is greatly affected by the tillage of soil, and covering with mulch will weaken the sensible heat exchange between the soil and the environment and block the moisture exchange between them, which can thus inhibit the latent heat exchange [65]. Moreover, covering mulch could promote the efficiency of crops to utilize light and heat, so that the cumulative temperature during the reproductive period of crops can be significantly increased, which will thus increase the crop yield. Covering agricultural mulch in arid areas can impede the vertical evaporation of water in the soil, thus promoting the horizontal transport of water and effectively conserving soil moisture. In some saline areas, covering the mulch can inhibit the return of saline salinity, so that it can form a special low-salt tillage layer under the ground, which can greatly reduce the hazards brought rendered by salinity [66]. In addition, covering the mulch can also inhibit the growth of weeds. Mulching can inhibit the growth of weeds and the occurrence of pests and diseases to increase crop production and prompt the rural economy.

Regarding lignin-based mulching, it not only poses the functions of traditional agriculture mulching (Figure 7), as mentioned above, but also has the advantages of biodegradability, renewability, and environmental friendliness. As shown in Figure 8, most reported lignin-based agriculture mulching generally has better tensile strength, Young’s modulus, and moisture retention than traditional PE mulch. In addition to the consideration of the low cost of lignin as an industrial waste, lignin-based composite film has displayed its potential to replace traditional PE film for agricultural mulching.

However, mulching for agriculture practice still faces some challenges. First, the practice of mulching can be impacted by the environment, such as wind, heavy rain, and other inclement weather. In these conditions, care should be taken to avoid the risk of tearing the film [71]. Second, color can enable different functions of mulching. The types of mulch to be selected for different crops are also different and need to be selected according to the crop growth characteristics [72,73]. Some biodegradable films can not be used for a long time, so farmers need to regularly replace the film to ensure the mulching is still functioning [74]. In addition, specific equipment is required for applying plastic mulch into the field, which requires the development of such mulching equipment [74].

#### 4.1.1. Current Status of Traditional Polyethylene (PE) Mulch Applications

The global market for plastic mulch is expanding dramatically across the globe as the population increases and the demand for food quantities grows every year, with around 6500 km^2^ of farmland currently using plastic mulch for global crop production [75]. The global plastic mulch market is estimated to reach USD 5.1 billion by 2027 [1]. In 2021, PlasticsEurope estimated that plastic film used in the European agricultural sector, including agriculture and horticulture, reached 1.02 metric tons, representing about 4% of the world plastics market [75]. In that year, China as the world’s largest film user, sowed nearly 17.4 million hectares of cover crops and used 1.45 million tons of mulch, representing more than 75% of the world’s total consumption [76]. In addition, in 2024, the North American agricultural film market size is expected to reach 1.24 billion U.S. dollars [77]. All these data indicated that global agriculture is increasingly reliant on mulching.

#### 4.1.2. Agricultural Surface Pollution from Traditional Agricultural Mulch

In recent years, with the continuous and profound development of science and technology, the thickness of agricultural mulching film is getting thinner and thinner. At the end of each cycle of production, agricultural films cannot be completely recycled, leaving macro-particle residues and microscopic plastic fragments. Studies have shown that low-density polyethylene (LDPE) films remain stable and difficult to degrade for decades [78]. If the residual film in the soil is not properly and effectively recycled and allowed to gradually accumulate, it will inevitably destroy the original physical and chemical structure of the soil, leading to a continuous decline in soil fertility, and ultimately affecting the deepening of the root system of the crops, reduce crop yields, and even lead to difficulties in the infiltration of groundwater, leading to salinization of the soil and other serious consequences [79].

Agricultural mulching film residue may hinder the connection between the soil and the air, leading to increased resistance to water movement than in ordinary soil. Therefore, water infiltration in the soil decreases with more film residue [80], resulting in a gradual decrease in soil water content. This weakens the drought resistance and negatively affects crop growth [80]. In the case of underground water sources, it is difficult to infiltrate and easy to trigger the soil’s secondary salinization, resulting in a large number of salt accumulation on the soil surface, thus greatly reducing the fertility of the soil [81].

Agricultural mulching film causes serious harm to the soil and will inevitably have a corresponding impact on the growth and development of crops. The film that has not been recycled will be entangled with the root system of plants [82]. With the increasing amount of residual film in the soil, the growth of crop roots will be inhibited and the residual film will be more difficult to remove, which will seriously affect the absorption of water and nutrients by crops, leading to poor crop growth and development, and in severe cases, even leading to yield loss and crop failure.

Traditional agricultural mulching film is mainly made of polyethylene and in order to increase the ductility of the film, 40–60% of plasticizers are generally added [83], which is toxic to the growth and development of crops, especially vegetables. In addition, most regions generally use collective incineration of residual films, the incineration process will produce dioxins, hydrogen chloride, and other toxic and harmful substances [84], causing serious harm to farmland, soil, and air quality.

### 4.2. Prospects for Lignin-Based Agricultural Mulching Film Applications

#### 4.2.1. Covering Performance of Lignin-Based Agricultural Mulching Film

The moisture retention ability of mulch is crucial in the practical application of crop production. The lignin-based agricultural mulching film has the same moisture retention properties as traditional plastic mulch [85]. Studies have shown that due to the physical cross-linking effect (hydrogen bonding) and coordination cross-linking between lignin and other materials, the structure of the film is denser [86], which is conducive to reducing the gas diffusion coefficient within the matrix, thus reducing water vapor evaporation from the soil and ensuring that the crop grows faster in the presence of a humid surrounding environment. For example, Wang et al. [87] prepared an environmentally friendly and structurally robust black composite film by compositing sodium lignosulfonate with sodium carboxymethylcellulose and polyvinyl alcohol, and the chelation and redox reaction between Fe^3+^ and sodium lignosulfonate consumed the polar oxygen-containing functional groups within the lignin to form a more stable three-dimensional structure, which improves the hydrophobicity of the film as well as its mechanical properties. The film prepared in this study was black in color, with good soil water retention, thermal insulation, controlled urea release, and good biodegradability.

In addition, there are also studies on the integration of pesticides, fertilizers, and agricultural films in one package in the form of liquid mulch film, which will become a new research hotspot. A representative study by Wang et al. [88] has polymerized lignin, starch, and acrylate monomers to prepare pseudo-plastic fluid mulch film with aerosolized properties. The crystallinity of starch was reduced by indirect branch copolymerization of lignin with starch and acrylate, thus improving the thermal stability and mechanical properties of the film. The covered mulching film not only promotes the germination rate of seedlings but also effectively maintains the temperature and moisture of the soil even after the seedlings have penetrated the soil and the mulch.

Furthermore, the addition of urea to the liquid mulch can not only serve as the fertilizer but also achieve its slow release, so that the crops can absorb more nitrogen during the growth process, thus promoting the growth of plants. Tian et al. [89] used water-soluble cyanoethyl lignin to prepare the liquid mulch, because its sprayed liquid mulch and soil particles can form a layer of film together, and the substrate itself organization and denseness of the structure will increase the length of the diffusion path of water molecules, which is conducive to reducing water evaporation. Overall, lignin agricultural mulch has the advantages of good UV-blocking, water resistance, soil consolidation, thermal stability, and good thermal insulation and entropy retention, all these advantages will benefit its covering performance as agriculture mulching.

#### 4.2.2. Biodegradability of Lignin-Based Agricultural Mulching Film

Agricultural production is still dominated by traditional non-biodegradable synthetic plastic mulch, most of which is made of low-density polyethylene and must be collected at the end of each growing cycle and sent to landfills for disposal. The processing of recycled plastic mulch is complex and costly, while mulch in agricultural fields is difficult to recycle completely due to crushing and other reasons, which can result in a series of hazards such as potential harm to microbial activity and crop growth due to toxic substances released from plastic residues in the soil [90].

Different from non-biodegradable synthetic plastic mulch, lignin-based agricultural mulch has excellent biodegradability and can be completely degraded by naturally occurring microorganisms such as fungi, bacteria, and actinomycetes into carbon dioxide and water, among others [91], without causing environmental pollution. In the degradation process of some bacteria, lignin is preferentially degraded than cellulose [92]. For instance, Su et al. [93] added quaternary ammonium lignin and sodium alginate together into polyvinyl alcohol to prepare the agricultural mulching film, in which the lignin incorporation improved the UV-resistance of the mulch and the synergism of sodium alginate improved the mechanical properties and water retention of the mulch.

In addition, in the degradation process, it was found that 55% of lignin-based agricultural mulch could degrade in 50 days when it was buried in the soil [93]. Therefore, the lignin-based mulch has been sought after as an excellent solution for replacing PE film to solve the white pollution problem. In addition, soil fungi can obtain the necessary nutrients while degrading lignin, so that they can replenish their own energy during the biodegradation of the film to further increase the degradation capacity. Such effects have been proved by Rodríguez et al. [94] that white rot fungi can degrade lignin well, thus promoting their own growth and later cleaning some soils that contain insecticides and pesticides.

#### 4.2.3. Enhancement of Soil Organic Carbon by Lignin-Based Agricultural Film

Different from traditional synthetic PE mulching, several microorganisms, including bacteria and fungi, can degrade lignin mulch and use it as a carbon source, which can eventually enable the return of lignin into the soil and solidify into organic carbon [95]. The enhancement of the organic carbon content in the soil is conducive to the enhancement of soil fertility, which improves crop productivity and promotes crop yield [63]. In addition, the slow-release effect of agricultural mulch can reduce the emission of greenhouse gases such as carbon dioxide and nitrogen oxides and promote the development of sustainable green agriculture. The conversion of lignin into agri-plastics also meets the need for agricultural carbon sequestration and the requirements for low-carbon and environmentally friendly development of agriculture.

Not only this, the conversion of lignin into mulching has the potential to alter the carbon footprint in the agriculture biosystem. Since lignin is a major component of crop cell walls, the fabrication of lignin into mulch could enable high-value utilization of crop straw, which will prompt the carbon flow and its efficiency in the agriculture ecosystem. At the end of the lignin-based mulching, it can remain in the field and convert into soil organic matter to better serve the planting for the next season, which could significantly improve the carbon footprint of agriculture.

Overall, the application of lignin-based agricultural mulching represents an effective pathway to reduce plastic pollution, enhance soil organic carbon, and increase carbon efficiency which could reduce greenhouse gas emissions during crop production to some extent.

#### 4.2.4. Economics of Lignin-Based Film Production and Application

Besides the advantages of lignin-based agriculture mulching, it could also have economic advantages. First, in terms of input cost, the price of lignin is relatively low compared to the cost of other biodegradable composites. PBAT, one of the most widely used biodegradable plastics, had a price as high as $4600/ton in early 2021 [96]. Lignin, on the other hand, was priced at around $440/ton in the same period [97], which was much lower than the price of PBAT. Second, the use of lignin-based mulching could increase crop yields and benefit rural economies. Compared with traditional mulching, lignin-based agricultural mulching could serve as a slow-release fertilizer after use and improve the fertility of the field since lignin itself is a soil organic matter [98]. Moreover, due to the unique internal conjugated structure of lignin itself, lignin-based agricultural mulch film can better convert light energy into heat energy and has better insulation properties of temperature and moisture [99], which is conducive to plant growth and development. Lignin agricultural land film also has good biodegradability, which makes it able to meet the needs of green agriculture as well as reduce the cost of traditional plastic mulch films [87]. For all these reasons, lignin-based mulching could be cost-effective and its application could have great potential to promote the rural economy.

## 5. Conclusions

Along with the continuous development of the world economy, the demand for food to feed the growing population is also increasing. The use of agricultural mulch has dramatically increased the yield of food, but the remaining plastic residuals derived from traditional plastic mulch have caused the destruction of soil structure and serious environmental pollution. Therefore, the development of biodegradable mulching such as lignin-based film can not only meet the increasing demand for food but can also protect the agriculture ecosystem prompting sustainable development of modern agriculture.

At present, the chemical structure and molecular weight of industrial lignin are uneven, which hinders the application of lignin-based film, such as the existence of impurities of sugars and ashes in industrially purified crude lignin can seriously affect the performance of lignin-based mulch. Recent global innovations focus on biodegradable mulch, among which, lignin-based agricultural mulch is paramount due to its cost-effectiveness together with excellent water resistance, mechanical properties, and covering performance which could doubtlessly prompt the sustainability of global agriculture.

## Figures and Tables

**Figure 1 polymers-16-02488-f001:**
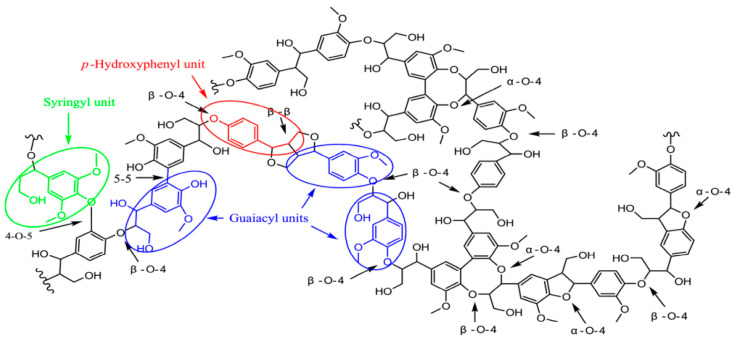
Typical lignin structures and intramolecular linkages [20]. The blue, green, and red colors in the figure represent the G-, S-, and H-type structures of lignin, respectively.

**Figure 3 polymers-16-02488-f003:**
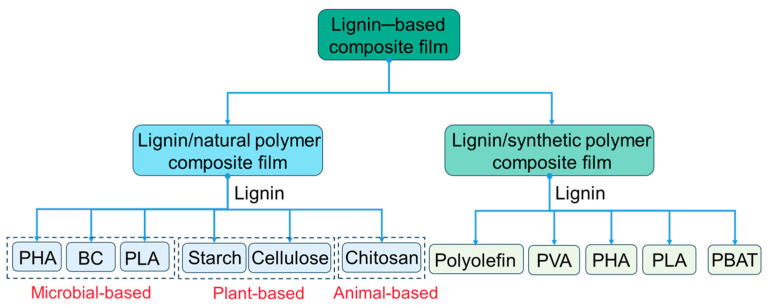
Classification of lignin-based compositive film. PHA: polyhydroxyalkanoates, BC: bacterial cellulose, PLA: polylactide, PVA: polyvinyl alcohol, PBAT: poly (butylene adipate-co-terephthalate).

**Figure 4 polymers-16-02488-f004:**
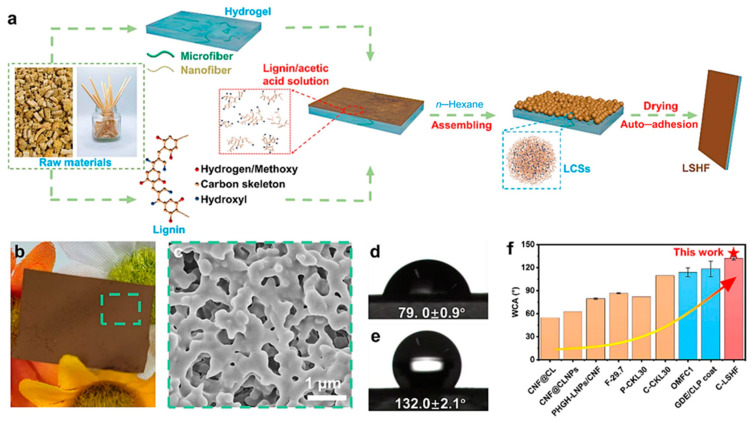
Preparation of hydrophobic cellulose/lignin composite film and its related performance characterization properties. (**a**) Hydrophobic cellulose/lignin composite film; (**b**) Image of hydrophobic cellulose/lignin composite film; (**c**) Surface morphology of hydrophobic cellulose/lignin composite film, and panel (**c**) is the magnified part of the dashed square in panel (**b**); (**d**,**e**) Water contact angle of lignocellulose film (**d**) and hydrophobic cellulose/lignin composite film (**e**); (**f**) Comparison of initial water contact angle (WCA) of hydrophobic cellulose/lignin composite film with other materials, and the arrow in panel (**f**) indicates the trend of the increased WCA [37].

**Figure 5 polymers-16-02488-f005:**
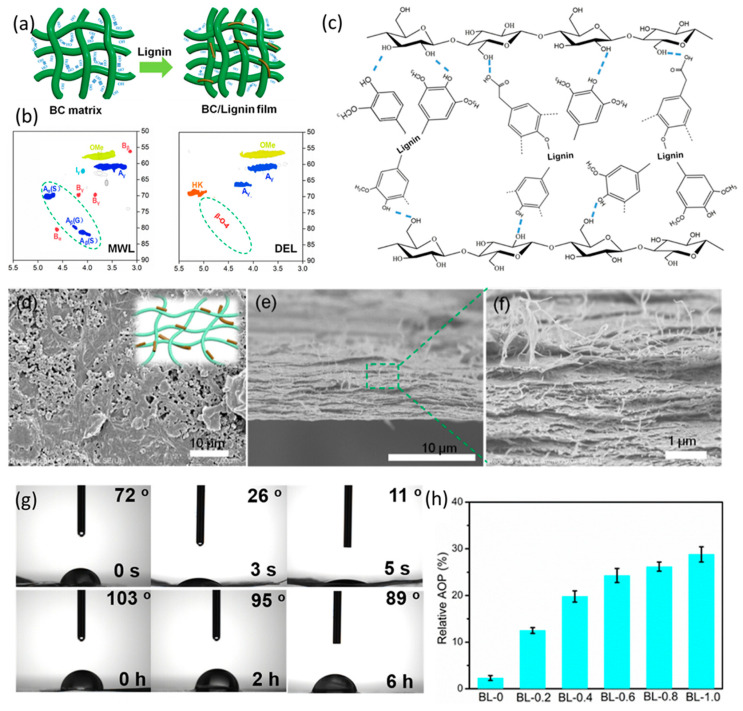
Preparation process of bacterial cellulose/lignin composite film materials and characterization of their relevant properties. (**a**) Design and preparation process of two films of lignin obtained by MWL and DES extraction; (**b**) schematic representation of 2D-HSQC NMR spectra, and the dashed lines emphasized the signals of the detected lignin structure; (**c**) schematic representation of cross-linking between lignin and bacterial cellulose obtained by DES extraction; (**d**) top-view SEM view of bacterial cellulose/lignin composite film; (**e**,**f**) cross-sectional SEM view of bacterial cellulose/lignin composite film, and panel (**f**) is the magnified part of the dashed square in panel (**e**); (**g**) cross-sectional SEM view of bacterial cellulose/lignin composite film water contact angle over time; (**h**) relative antioxidant activity of films with different lignin contents [50].

**Figure 6 polymers-16-02488-f006:**
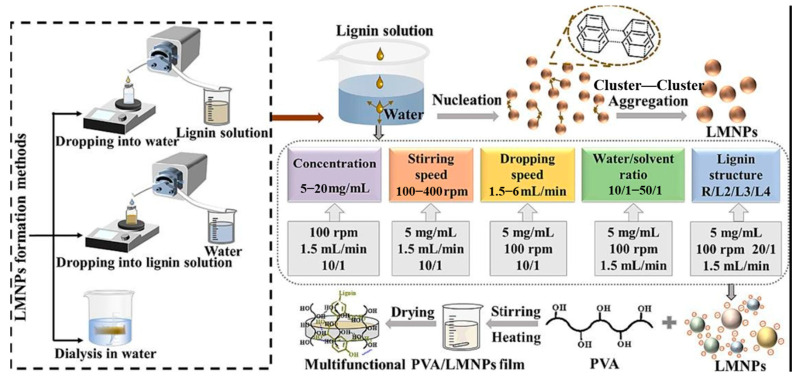
Green preparation of lignin nanoparticles and polyvinyl alcohol/lignin nanoparticles composite film [57].

**Figure 7 polymers-16-02488-f007:**
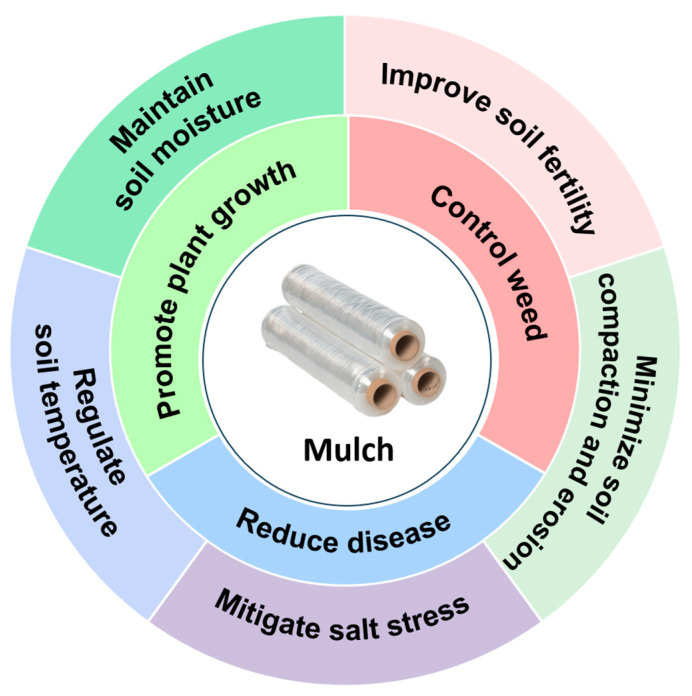
Functions of agricultural mulching.

**Figure 8 polymers-16-02488-f008:**
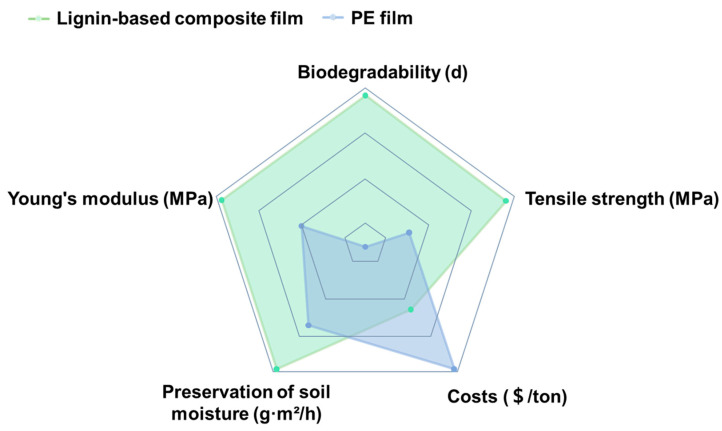
Comparison of properties between lignin-based composite film and PE films [67,68,69,70].

**Table 1 polymers-16-02488-t001:** Comparison of different parameters between lignin/natural polymer composite film and lignin/synthetic polymer composite film.

Classification	Composite Polymers	Lignin Source	Interfacial Interaction between Polymers	Function of Lignin	Performance	Ref.
Lignin/natural polymer composite film	Lignin/cellulose composite film	Acetic acid lignin	Physical adhesion between ester groups (C=O, O-C=O)	Self-adhesion	Highly hydrophobic film with tensile strength of 37.7 MPa	[37]
	Lignin/starch composite film	Lignosulfonate	Hydrogen bonding	Plasticizers and surfactants	Highly standard plasticized film with an elongation at break of 208%, Water content of 13.6%	[41]
	Lignin/chitosan composite film	Lignin	Hydrogen bonding and van der Waals forces	Provides phenolic groups and a high density of -OH groups	Recyclable film with adsorption capacity, tensile strength of 41.45 MPa	[47]
	Lignin/bacterial cellulose composite film	Lignin	Lignin attached to the reticulum of bacterial cellulose and hydrogen bonding	Retard free radical-induced oxidation, enhance mechanical property, and UV-blocking	Stronger interfacially bonded films, tensile strength of 343 MPa	[50]
Lignin/synthetic polymer composite film	Lignin/polyolefin composite film	Hardwood lignin modified by esterification-linear low-density polyethylene	Hydrogen bonding and physically bonded interfacial interactions	Enhance mechanical property, thermal stability and antioxidants, UV absorption	Flexible film, elongation at break of 351.33%, opacity of 46.83%	[55]
	Lignin/PVA composite film	Lignin micro/nanoparticles	Hydrogen bonding and polar bonding	Improvement of water absorption and heat resistance of composites	High crystallinity film with tensile strength of 99.4 MPa, resilience of 97.1 MJ/m^3^	[57]
	Lignin/PLA composite film	Lignin in corn stover	Heterogeneous crystallization	Nucleating agent, plasticizer	High toughness film with Young’s modulus of 1589 MPa, tensile strength of 55.1 MPa	[58]
	Lignin/PBAT composite film	Alkaline soda lignin	Intermolecular hydrogen bonding	Enhance mechanical property, thermal stability, and UV absorption	Films with excellent UV shielding properties, elongation at break of 689%, Young’s modulus of 63 MPa, tensile strength of 30 MPa	[59]
	Lignin/PHA composite film	Grape seeds lignin	Interfacial adhesion	Antioxidant and nucleating agent	Highly antioxidant film with E-modulus of 827 MPa, tensile strength of 82 MPa	[60]

## Data Availability

No new data and resources were produced to complete this review.

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
