# Peer review of "Lignin-Based Composite Film and Its Application for Agricultural Mulching"

_polymers, 2024, doi:10.3390/polym16172488_

Round 1

Reviewer 1 Report

Comments and Suggestions for Authors

Section

Line number (Page no)

Comments

Abstract

9-10 (Page 1)

Why mention only China and not any other countries on the globe as a whole? Polymer is an international journal, so globally it should be addressed or the title of the manuscript should be specific to agriculture in China.

1. Introduction

22 (Page 1)

Please include Modern and conservation agriculture

29 (Page 1)

Add plant/ crop biomass and not just biomass.

22 (Page 1)

Please enlighten the readers with the necessities of mulching in agriculture (include soil and water conservation, weed management, nutrient management aspects of mulching).

27 (Page 1)

Please define white pollution to make it easy and clear to the audience

26-27 (Page 1)

Include all the adverse effects of using plastic mulch in agriculture (include the adverse effects of the PE microplastic on soil, plants, humans, and the environment in brief). Please refer:  

Garai, S., Bhattacharjee, C., Sarkar, S., Moulick, D., Dey, S., Jana, S., ... & Hossain, A. (2024). Microplastics in the soil–water–food nexus: Inclusive insight into global research findings. Science of The Total Environment, 173891.

(Page 1)

Provide a graph showing the distribution of types of various mulching materials (plastic, paddy straw, organic, etc.) to depict the increasing threat of plastic as a mulching material in different countries

3. Lignin-based composite film materials

78-85 (Page2-3)

Add citation to the first paragraph

189 (Page 5)

Add citation

206 (Page 5)

Add citation

248 (Page 7)

Add citation

Add a flowchart to show the classification of Lignin-based compositive film

Add a table to compare the various parameters between natural and synthetic lignin-based film.

4. Application of lignin-based composite film in agriculture

297-299 (Page 8)

Add citations for the data

301 (Page 8)

Use “Regulate soil temperature “in place of “increase soil temperature” as the organic mulch increases soil temperature during winter and vice versa during summer.

304 (Page 8)

Add that organic mulch aids in nutrient management by releasing essential plant nutrients and organic carbon upon gradual decomposition by microbes.

(Page 8)

If possible, add a diagram showing all the advantages of mulching in agriculture.

(Page 8)

Add some constraints of mulching if present.

4.1.1. Lignin/polyolefin composite film

326-328 (Page 8)

Replace the old data with the most recent one.

330 and 334 (Page 9)

Why only China?

4.1.2. Lignin/polyolefin composite film

348-355 (Page 9)

Add citation

4.2. Prospects for lignin-based agricultural mulching film applications

(Page 9)

Add a comprehensive table depicting the research results of  effect of lignin-based mulch in yield increment in various field and horticultural crops.

A section mentioning the economics of production and application of lignin-based mulch should be added to complete the review.

Add a table showing the results on the effect of lignin-based mulch on the soil microbes.

4.2.1. Covering performance of lignin-based agricultural mulching film

372-373 (Page 9)

Add citation

4.2.3. Enhancement of soil organic carbon by lignin-based agricultural film

434 (page 11)

Add citation

Mention the carbon footprint in production of lignin based mulch as compared to plastic mulch.

5. Conclusion

439 and 451 (Page 11)

Why only China but, not globally? Please modify accordingly. 

Comments on the Quality of English Language

English language is more or less fine. Some modification is required. 

Reviewer 2 Report

Comments and Suggestions for Authors

Dear Authors,

I deeply reviewed the paper entitled "Lignin-based Composite Film and its Application for Agricultural Mulching." The topic is indeed interesting and holds significant potential for sustainable agricultural practices. However, there are several areas where the paper could be improved to enhance its overall impact and clarity.

1. Abstract: The abstract provides a concise summary of the paper; however, it limits the applicability of the research by focusing solely on China. Given the global relevance of sustainable agricultural practices, I suggest broadening the scope to emphasize the worldwide potential of lignin-based composite films. The topic is not exclusive to China and should be framed as a global solution, applicable in various agricultural contexts across different regions.

2. Introduction: The introduction is currently too brief and lacks sufficient justification for the selection of lignin as the primary biomolecule. While lignin is indeed an important material, the authors should provide a comparison with other potential biomolecules, such as cellulose or keratin, to establish why lignin is the most suitable choice for agricultural mulching applications. A more detailed rationale would strengthen the foundation of the review and better situate it within a global context.

3. Section 2.2: Lignin Separation Techniques: This section would benefit greatly from the inclusion of visual aids, such as figures or diagrams, to illustrate the various lignin separation techniques discussed. Visual representations from relevant literature would enhance the reader's understanding and provide a clearer picture of the processes involved. Additionally, these illustrations can help to highlight any challenges or advantages associated with each technique.

4. Comparative Analysis: The review currently lacks a thorough comparative analysis of lignin-based composite films with other materials used for similar applications. It would be beneficial to include a discussion comparing the physico-chemical properties, efficiency, cost, durability, and other relevant factors of lignin-based films with non-natural materials. This critical comparison would provide a more comprehensive view of the advantages and limitations of using lignin in agricultural mulching.

5. Global Perspective: As mentioned earlier, the review should expand its focus from a regional (Chinese) perspective to a global one. The potential applications of lignin-based composite films are not confined to a single country or region; therefore, the review should discuss the broader implications and benefits of this technology on a global scale. This would not only increase the relevance of the paper but also make it more impactful.

Round 2

Reviewer 1 Report

Comments and Suggestions for Authors

Authors has revised the manuscript significantly by resolving all my comments. In this present from, the manuscript may be accepted for publication. 

Comments on the Quality of English Language

English language is fine and satisfactory. Minor editing of English language required.

Reviewer 2 Report

Comments and Suggestions for Authors

Dear Authors, I have deeply reviewed the revised version of your manuscript, and I am pleased to acknowledge the significant efforts you have made in addressing my previous comments and recommendations. The improvements in the manuscript are evident, and the clarity and scientific rigor have been notably enhanced.

In my opinion, this paper could be considered for publication after these modifications.